# Loop-Mediated Isothermal Amplification of DNA (LAMP) as an Alternative Method for Determining Bacteria in Wound Infections

**DOI:** 10.3390/ijms25010411

**Published:** 2023-12-28

**Authors:** Monika Gieroń, Paulina Żarnowiec, Katarzyna Zegadło, Dawid Gmiter, Grzegorz Czerwonka, Wiesław Kaca, Beata Kręcisz

**Affiliations:** 1Faculty of Medicine, Jan Kochanowski University in Kielce, 25-369 Kielce, Poland; monika.chlon@gmail.com (M.G.); beata.krecisz@ujk.edu.pl (B.K.); 2Dermatology Department, Provincial General Hospital, 25-317 Kielce, Poland; 3Department of Microbiology, Institute of Biology, Jan Kochanowski University in Kielce, 25-406 Kielce, Poland; pzarnowiec@ujk.edu.pl (P.Ż.); kasienka977-97@wp.pl (K.Z.); dgmiter@ujk.edu.pl (D.G.); wkaca@ujk.edu.pl (W.K.)

**Keywords:** chronic wound, bacterial infection, rapid detection, genotyping, qPCR, LAMP

## Abstract

The increasing number of patients with chronic wounds requires the development of quick and accurate diagnostics methods. One of the key and challenging aspects of treating ulcers is to control wound infection. Early detection of infection is essential for the application of suitable treatment methods, such as systemic antibiotics or other antimicrobial agents. Clinically, the most frequently used method for detecting microorganisms in wounds is through a swab and culture on appropriate media. This test has major limitations, such as the long bacterial growth time and the selectivity of bacterial growth. This article presents an overview of molecular methods for detecting bacteria in wounds, including real-time polymerase chain reaction (rtPCR), quantitative polymerase chain reaction (qPCR), genotyping, next-generation sequencing (NGS), and loop-mediated isothermal amplification (LAMP). We focus on the LAMP method, which has not yet been widely used to detect bacteria in wounds, but it is an interesting alternative to conventional detection methods. LAMP does not require additional complicated equipment and provides the fastest detection time for microorganisms (approx. 30 min reaction). It also allows the use of many pairs of primers in one reaction and determination of up to 15 organisms in one sample. Isothermal amplification of DNA is currently the easiest and most economical method for microbial detection in wound infection. Direct visualization of the reaction with dyes, along with omitting DNA isolation, has increased the potential use of this method.

## 1. Introduction

The treatment of chronic wounds, even with the rapid progression of medical techniques, remains demanding. Chronic wounds have become epidemic. Every day, many clinicians struggle with the selection of appropriate methods for the diagnosis and treatment of chronic wounds [1]. A sedentary lifestyle, low physical activity, and poor eating habits are the cause of civilization-related diseases such as obesity, cardiovascular disease, atherosclerosis, diabetes mellitus, and chronic venous insufficiency, which disturb the normal healing process and promote the formation of nonhealing wounds [2]. The increasing number of patients with chronic wounds requires the development of quick and effective diagnostics and treatment methods. One of the key and challenging aspects of the treatment of chronic ulcers is controlling the infection. Bacteria can exist in two different states: the planktonic state, in which the cells are free-floating, and the sessile state, in which the cells are adhered to a surface, also called biofilm [3,4,5,6,7]. Biofilms are an organized network of many different species of bacteria and yeasts and other substances such as extracellular DNA or proteins. These microorganisms and substances are attached to the wound bed. This form of colonization increases bacterial survival in the wound environment by developing a network of mutual dependencies and increasing virulence [8,9,10,11]. The coexistence of two or more bacterial species makes them more resistant to antibiotics, a phenomenon known as antimicrobial synergism [12]. The phenomenon is best described for *Staphylococcus aureus* and *Pseudomonas aeruginosa*, which are the most common pathogens in chronic wounds [13,14]. Biofilm formation is an important mechanism underlying the observed delayed healing and is a major problem in chronic wound management [15]. Extensive data exist that substantiate the interference of microbes with wound healing. The reduction of microbial bioburden is essential for the ongoing process of tissue repair. Both systemic and topical antimicrobials play a crucial role in decreasing bioburden and promoting wound repair [16,17]. Early detection of infection is essential for the application of suitable treatment methods. The diagnosis of wound bed infection is made by the presence of symptoms, including odour, pain, purulent exudate, warmth, tenderness, or erythema and abnormalities in laboratory test results, such as elevated CRP, ESR, or procalcitonin [18,19,20]. In acute wounds these signs are usually obvious, but in the case of chronic wounds, which are colonized by many strains of bacteria, it is sometimes difficult to clearly determine when the infection has progressed. This is usually based on the development of acute symptoms or other signs, such as deterioration of the wound, an increase in wound size, bridging of the epithelium or soft tissue, formation of pockets at the base of the wound, purulent exudate, the appearance of friable granulation tissue which bleeds easily, abnormal smell, unexpected pain or tenderness, failure to make satisfactory progress toward healing despite optimal care [21,22]. Infection, not treated quickly enough, may result in further spread of infection and development of cellulitis, fasciitis, and myositis, and may finally lead to bacteremia and septic shock. Therefore, rapid assessment of the pathogens causing the infection is key to implementing targeted antibiotic therapy [21]. The problem is more complex due to limitations of the commonly used method of identifying bacterial strains colonizing the wound. In general practice, the most used, and sometimes the only available, method to diagnose wound infections is through a bacterial swab from the wound, followed by culture on appropriate bacterial media [23]. The biggest limitation of this technique is the long detection time. Culture is performed by inoculating an appropriate medium with the bacterial swab and incubating it for a period of time. Most clinical pathogens grow from 24 to 48 h, but certain isolates sometimes require longer incubation periods, i.e., 2 or 3 weeks for *Bartonella* spp. or *Nocardia* spp. infections [23]. Then, the plates are examined and then reincubated for additional time if no growth is observed. The overall process takes 1 to 7 days, depending on the type of bacteria, the medium, and the growth conditions [23,24,25]. This delay in detection has a negative impact on the implementation of treatment for patients with chronic wounds. In the case of hospitalization of patients with acute or severe chronic wounds, the usage of a quick detection method as an indicator of the microbiological condition of the wound would shorten the hospitalization time through early administration of targeted antibiotic therapy. This would reduce the overall treatment costs.

Moreover, the results depend on the method of collecting the wound swab. When the procedure is done incorrectly, or too superficially, it may result in false outcomes [26,27,28]. While there are a few collection methods used in general practice, there is no consensus on which is the most accurate [23,25,27]. The viability of organisms also depends on several factors, such as the transport time, storage period, and temperature, as well as the specific storage systems [18]. Prompt delivery of the specimen to the laboratory is also important, particularly if anaerobic bacteria are being investigated [25]. In several published studies comparing culture to molecular methods for the detection of bacteria, molecular methods were more sensitive. Most bacteria identified with culture were also identified via molecular testing, but the vast majority of bacteria identified via molecular methods were not identified with culture methods [28,29,30,31,32]. Even if all culture steps are performed correctly, many anaerobic bacteria, which constitute up to 30% of the bacteria infecting wounds [33], detected by molecular methods cannot be cultured [23]. These results translate into clinical practice; targeted personalized therapy, implemented through molecular diagnostics methods, can increase healing rates and shorten healing time compared to conventional methods [34,35].

All of the issues associated with the use of conventional diagnostic methods have generated interest in the use of other methods to diagnose infections. In this article, we present selected molecular methods that can be used for the rapid detection of pathogens causing wound infections. We focus on the LAMP method, which has not yet been widely used to detect bacteria in chronic wounds but is an interesting alternative to conventional detection methods. The most common and available methods are the real-time polymerase chain reaction (rtPCR) and quantitative polymerase chain reaction (qPCR), which allow rapid assessments of the presence of microorganisms in a sample. PCR is sensitive and does not require a large amount of material to amplify nucleotide sequences. Genotyping using random primers with template DNA is another diagnostic method that allows us to distinguish between microorganisms that cause infections. Loop-mediated isothermal amplification (LAMP) is an alternative detection method that does not require additional equipment and provides the fastest detection time for microorganisms. In LAMP, many pairs of primers can be used in one reaction, and 15 organisms can be detected in a single sample.

## 2. Real-Time Polymerase Chain Reaction (Real-Time PCR, qPCR)

The polymerase chain reaction (PCR) was developed in 1983 by Mullis [36]. This method quickly contributed to significant progress in many fields of science and in diagnostics. The method is based on the amplification of DNA by a thermostable DNA polymerase by repeated heating and cooling of the sample under controlled conditions. The initiation of amplification of a fragment of genetic material requires the presence of primers, i.e., short fragments of single-stranded DNA, that bind to the complementary sequence. This allows a specific selection of the duplicated fragment [37].

Numerous modifications to the method have been developed over the years, including real-time PCR (quantitative PCR, qPCR), which allows qualification of the abundance of amplified fragments in real time. This is due to the use of fluorescent dyes or fluorochrome-labelled molecular probes. The level of fluorescence emitted by molecular probes or intercalating dyes is monitored and is directly proportional to the concentration of the amplified DNA. Changes in fluorescence are presented in the form of a curve of fluorescence intensity. In the initial phase, DNA amplification is slow, which is due to the low level of fluorescence emissions, recorded as background. The cycle in which the fluorescence exceeds the background level is called the cycle threshold (Ct). The time required to achieve the Ct is inversely proportional to the initial concentration of the DNA template in the sample. Determination of the cycle threshold in many samples allows a comparison of the number of copies of the DNA fragment amplified by the primers [38,39].

Real-time PCR has been widely used for the detection and differentiation of microorganisms from various samples, both environmental and clinical, since the method allows amplification of a specific fragment of DNA [36]. For example, based on qPCR, Trung and colleagues developed a method for direct detection and quantification of *Burkholderia pseudomallei*. Bacterial identification is based on the targeting of a type-three secretion system 1 single-copy gene [40]. Moreover, a method has also been developed for detecting *Campylobacter* bacteria in environmental samples, which is important from a clinical point of view. For this method, strain-specific fluorescent-labelled PCR oligonucleotide probes were designed based on the short variable region of the *flaA* gene [41]. This method might be adopted for the detection of *Campylobacter* in clinically relevant samples.

In terms of methods designed for the detection of bacteria in clinical samples, Chiba and colleagues reported an identification system using real-time PCR with pathogen-specific molecular beacon (MB) probes and primers designed for eight meningitis pathogens [42]. In another study conducted by Anbazhagan and colleagues, a protocol for the identification of important nosocomial pathogens (e.g., *Escherichia coli*, *Staphylococcus aureus*, *Streptococcus pneumoniae*, *Klebsiella pneumoniae*, *Pseudomonas aeruginosa*, and *Acinetobacter* spp.) was developed. The specificity of this multiplex PCR assay was determined with 300 clinical samples [43]. Curran and colleagues developed and evaluated real-time PCR assays for simultaneous, direct detection and quantification of a range of respiratory bacteria in individuals with chronic obstructive pulmonary disease. They revealed that the prevalence of bacteria detected by real-time PCR compared to that detected by culture was substantially higher for *Streptococcus pneumoniae*, *S. aureus*, *Haemophilus* spp., and *Moraxella catarrhalis* [44]. Zemanick and colleagues described a qPCR-based protocol for the detection of *P. aeruginosa, S. aureus*, and *Haemophilus influenzae*, common pathogens in cystic fibrosis (CF) patients. These authors found that qPCR is a reproducible method for the detection of bacteria, including anaerobic bacteria, from CF airway samples [45]. Gosiewski and colleagues proposed qPCR-based methods for bacteria and fungi detection in blood collected from patients with clinical symptoms of sepsis. They focused on *E. coli*, *S. aureus*, *Candida albicans*, and *Aspergillus fumigatus*. Their results indicated that the methods (1) allow the detection of bacteria in whole blood samples, (2) are more sensitive than culture-based methods, and (3) allow the differentiation of the main groups of microorganisms within a few hours [46]. Gaibani and colleagues also helped develop methods for bacteria detection in blood and plasma. They described a broad-range real-time PCR protocol targeting the 23S rDNA gene in DNA extracted from plasma and whole blood samples [47].

Considering the methods of detection of bacteria in chronic wounds, studies have revealed that molecular methods have high potential for the diagnosis of bacterial infections [31,48]. Such methods have shown similar or even higher accuracy and sensitivity than traditional culture-based techniques and are faster and simpler. However, there are limited reports on the use of qPCR for the identification of bacteria isolated from wound infections. Melendez and colleagues developed a suite of real-time PCR assays for the rapid identification of bacteria directly from tissue samples. They proposed methods of identification based on a panel of 14 clinically relevant aerobic pathogens in chronic wounds. The methods were based on primers and molecular probes that target and amplify a fragment of the 16S rRNA gene sequence. They demonstrated that a targeted real-time PCR approach can be used for the rapid detection of aerobic organisms isolated from chronic wounds [49]. In the study by Gentili and colleagues, a panbacterial quantitative real-time PCR method was tested to evaluate its potential in the diagnosis of wounds treated with a novel therapeutic approach based on the hydrophobic binding of bacteria to a membrane [50]. Finally, Yan and colleagues presented a slightly different approach that allows the detection of *Mycobacterium leprae* DNA in paraffin-embedded skin biopsy specimens using real-time PCR [51]. qPCR could also be used for the detection of antibiotic resistance genes in numerous samples [52,53,54]. In the era of increasing drug resistance among bacteria, developing rapid methods for the detection of bacteria containing resistance genes is important, as it will facilitate the introduction of appropriate treatment regimens.

## 3. Genotyping of Strains from Wound Infections

Molecular typing has evolved into a pivotal approach for unravelling the genetic diversity among bacterial isolates and understanding the mechanisms underlying bacterial infections in hospital settings, including those from ulcerative wounds. It plays a crucial role in investigating the spread, clonal relationships, and geographic dissemination of bacterial strains, which is crucial for managing cross-infections and patient-to-patient transmission in healthcare environments. By enabling precise identification of pathogens and their resistance profiles, molecular typing facilitates the formulation of targeted antibiotic regimens, crucial for treating chronic, non-healing ulcers, often complicated by biofilm and polymicrobial flora. In the realm of infection control, molecular typing methods stand out as essential tools for measuring and pinpointing the source of original infections during hospital outbreaks. Furthermore, they enable the mapping of infection transmission dynamics, offering a deeper insight into the epidemiology of antibiotic-resistant strains. This approach not only informs the debridement techniques and wound dressing choices, optimizing the wound microenvironment for healing, but also underpins the development of prophylactic measures against recurrent ulcer infections. Understanding the genetic evolution of pathogens through molecular typing also helps predict emerging resistance trends, guiding future antimicrobial development, and contributing to a comprehensive strategy in managing and preventing ulcerative wound infections. Thus, molecular typing is not merely a diagnostic tool but a cornerstone in the strategic management of infections, especially in ulcerative wounds, enhancing patient care and advancing the field of clinical microbiology.

Genetic typing techniques are characterized by a wide variety of technical solutions. Among the most commonly used molecular typing methods are techniques such as PCR-based fingerprinting methods, multilocus sequence typing (MLST) [55], and pulsed field gel electrophoresis (PFGE), the gold standard in genotyping [56]. MLST allows for the differentiation of bacterial strains based on the sequence of selected housekeeping genes [55]. In this method, 7 to 10 genes, each 450–500 bp, are amplified in a PCR reaction. Due to their conservatism, these genes have been recognized as reference genes in genetic studies [57]. They are responsible for the proper course of the most essential life processes of organisms, and their elimination always leads to cell death. The proteins encoded by these genes are involved, among other things, in processes such as replication or translation [58]. In the next step, the PCR products are sequenced, and the obtained sequences are compared to alleles for each locus of the gene using software available in the MLST database. Then, for each isolate, an allelic profile is generated by determining the sequence type. The frequent use of this technique in genetic studies is evidenced by the existence of publicly available reference databases of MLST profiles for many different bacterial species [59]. The greatest advantage of the past years is the culture-independent application of multilocus sequence typing (MLST) for strains like *Pseudomonas aeruginosa* and *Burhholderia cepacia*. This process was performed on DNA extracted directly from patient clinical material [59,60]. In contrast, PCR-based fingerprinting methods, such as repetitive extragenic palindromic (REP) sequences, the enterobacterial repetitive intergenic consensus (ERIC) sequence, and BOX elements, random amplified polymorphic DNA (RAPD), and arbitrarily primed PCR (AP-PCR) offer a more cost-effective and user-friendly alternative [61]. One of the most popular is the RAPD PCR technique developed by Williams and colleagues [62]. The basis of the method relies on the random amplification in a simple PCR reaction using primers with any, often random, sequence of 10 to 20 bp, containing a large number of guanine (G) and cytosine (C) bases. In the initial cycles, the reaction is conducted at a lower temperature (approximately 35–40 °C), allowing the primers to randomly attach to more or less homologous sequences of bacterial DNA, leading to the amplification of DNA between primer binding sites. Subsequent cycles are performed at a higher temperature, suitable for the selected primer. As a result of amplification, several to several dozen amplicons are obtained in the range of approximately 100–2000 bp. The outcome of the PCR reaction is a specific genotypic profile. The obtained band profile reflects differences in the nucleotide sequence of primer binding sites, variations in the length of amplified fragments, and the DNA conformation affecting efficient primer binding and amplification. The smaller the relatedness between the examined strains, the greater the differences in the number and length of amplicons [63,64].

RAPD patterns are commonly used for strain typing, allowing for the analysis and comparison of isolates. Due to its simplicity, high discrimination power, relative speed, and ease of execution, the RAPD technique has found wide application in epidemiological studies. However, RAPD PCR has limitations, as band patterns are not always repeatable when the method is not optimized and standardized. Differences in the type and quantity of DNA polymerase, buffer composition for PCR, purity and quantity of DNA template, PCR reaction parameters, and the nonspecific binding of primers can lead to changes in RAPD patterns, resulting in low reproducibility of results obtained in different laboratories. Additionally, this technique does not always allow for the differentiation of closely related strains. Currently, the RAPD method is used for the preliminary diagnosis of bacterial infections or as an auxiliary method confirming results obtained by other bacterial strain typing methods [65,66]. PCR-based approaches are fast and effective methods for quantifying the number of genes and/or transcripts in the samples examined. They have high specificity and sensitivity towards target sequences. However, to maximize the value of the result based on RT-PCR, it is worth using it in combination with other methods [67].

A method of identifying microorganisms using next-generation sequencing of the entire bacterial genome is currently attracting much interest. It is an extension of the Sanger sequencing method and is becoming faster, more accurate and cheaper, which will allow it to be used routinely in clinical microbiology in the future [68]. Next-generation sequencing consists of four main steps:-Preparing the sample and isolating the genetic material;-Creation of libraries that will be compatible with the selected sequencer;-Amplification and the sequencing process;-Computer processing of the obtained data and its analysis [69].

There are two main sequencing technologies: “second-generation”, or short reads (Illumina and Ion Torrent) and “third-generation”, or long reads (Pacific Biosciences and Oxford Nanopore) [70]. For bacterial identification, 16S rDNA sequencing is particularly important for bacteria with unusual phenotypic profiles, rare bacteria, slow-growing bacteria, uncultivable bacteria, and culture-negative infections [71]. The next-generation sequencing used in this study is beginning to provide valuable information about the composition, diversity, and dynamics of the wound’s bioburden, but also indicates the existence of a link between it and impaired healing, and the possibility of infectious complications [72]. The current sequencing of the skin microbiome is crucial for the future development and application of diagnostic tools linking the microbiome to chronic wounds and focusing on the dynamic metatranscriptome and metaproteome [73]. The biggest advantages of using sequencing in diagnostics are its accuracy, comprehensiveness, and the large amount of information on the genome under study, while the main disadvantages are the possibility of missing species due to primer mismatches, the necessity of performing bioinformatics analyses (often complex and sophisticated) requiring access to databases, the length of the entire process, and the availability of specialized equipment (sequencer) [74].

Despite the continuous development of molecular methods, especially whole-genome sequencing, genotyping methods are still used in the identification of bacteria in wounds. Particularly, in screening or epidemiological studies, when dealing with a large number of strains, the cost of whole-genome sequencing (WGS) may be too high. Additionally, in studies of clonal sources of infection, the ST number remains an international unit [75,76,77,78,79,80].

## 4. Loop-Mediated Isothermal Amplification of DNA (LAMP)

Loop-mediated isothermal amplification of DNA, developed by Notomi and colleagues in 2000 [81], allows for rapid and accurate diagnoses in 30 min. it employs a set of four to six primers that reproduce the high selectivity of target sequence detection. LAMP requires a polymerase with strand displacement activity (*Bst* DNA polymerase, derived from *Geobacillus stearothermophilus*), and amplification relies on auto-cycling strand displacement DNA synthesis in isothermal conditions. Primer design is a crucial step in the optimization of the LAMP reaction; two pairs of primers recognizing six independent sequences of a target gene and additional forward loop primers that accelerate the LAMP reaction must be designed [82]. Visualization of the results can include a visual assessment of turbidity, discrimination of dye colour change, or measurement of the fluorescence of reagents, such as ethidium bromide or SYBR Green I [81,83,84].

### 4.1. Strand Displacement Synthesis

The strand displacement activity of polymerases occurs during the removal and replacement of RNA primer moieties of Okazaki. The strand displacement activity is present in Klenow fragments [85]—a polymerase derived from digestion of polymerase I (PolI) by subtilisin; the 5′-nuclease activity is not necessary for strand displacement synthesis [86]. Strand displacement DNA synthesis is performed based on usage of a forward primer and a DNA polymerase with strand displacement activity. With primer extension, the new synthetic DNA strand can displace and release the downstream complementary strand [87]. Moreover, intact double-strand DNA molecules cannot be used as a template for polymerase in strand displacement synthesis, but when ssDNA breaks occur (a nick) with an extensible 3′-hydroxyl termini on the DNA strand, synthesis occurs. In strand displacement synthesis, DNA polymerase I starts to displace the 5′ end of the nick, which is annealed to the template strand (Figure 1). However, strand displacement synthesis under isothermal conditions often leads to nonspecific amplification products, which can be obtained even in the absence of a target and primers [88]. Experimental results indicate that non-specific isothermal polymerization may be generated by a stronger interaction between polymerase and purine-rich regions of a single stranded DNA template than in double-stranded primer-tagged regions [89].

### 4.2. Mechanism of LAMP Reaction

Isothermal amplification of DNA using LAMP requires four or six primers that recognize 6–8 DNA regions. These primers include external primers (forward primer—F3, backward primer—B3), long internal primers (forward internal primer—FIP, backward internal primer—BIP), and loop primers (FL, BL) [91]. The long internal primer sequences, which are approximately 45–49 base pairs (bp), are complementary to two distant sites of the area in which DNA synthesis is initiated. External primers are complementary to external modules, trigger strand replacement, and are shorter than the internal primers (ca. 21–24 bp) [92]. Proper design of primers is a crucial step, especially for internal primers, where modification of the linker, which is expected to provide physical flexibility, can result in sequence interactions that can alter reaction efficiency [93]. Synthesis of the complementary strand in the LAMP reaction is initiated by attachment to the F2 complementary region in the template strand of the internal FIP primer in the DNA template [94]. Subsequently, DNA strand displacement and release occur with the attachment of the external primer F3 to the target DNA region. The released single strand combined with FIB forms a stem-and-loop structure at one end, which initiates BIP-mediated DNA synthesis [95]. The reaction uses polymerase *Bst*, which localizes external primers and is responsible for elongating strands and forming single-stranded DNA, which is a dumbbell-shaped artificial template [92] (Figure 2). Internal primers are then bound to the stem-loop region, and sequentially repeated strand elongation reactions occur via polymerase [96]. By annealing between alternatively inverted repeats of the target in the same strand, cauliflower- or dumbbell-like structures are formed, which, together with stem-loop DNA structures of different lengths, constitute the mixture of end products of the LAMP reaction [97].

### 4.3. Diagnostic Use of LAMP

When there is a need to design and develop on-site molecular diagnostics, so-called ‘Point-of-Care tests’ [99], LAMP is of interest to the scientific community because of its advantages over PCR/qPCR in applications such as lab-on-chip device development or point-of-care testing. Isothermal amplification techniques do not rely on expensive equipment and in many cases can be combined with various strategies for interpreting results with the naked eye, making this step user-friendly [93]. The advantages of LAMP include (i) equipment simplicity: the reaction requires only a heating block or water bath [100], (ii) short time from sample collection to results: the use of dyes shortens diagnostics to 70–120 min [84], specificity, and high amplification efficiency [101].

Untreated wound infections can rapidly evolve into sepsis and fast and simple detection of the microorganism that caused the infection is crucial for diagnosis and treatment. Simple and rapid LAMP, where visualization is based on a colour change, was engaged to detect 15 species of common microbial pathogens in studies by Etchebarne and colleagues [84]. LAMP amplification is especially advantageous because of its short detection time, without prior DNA isolation. Amplification was performed, for the first time, on DNA templates obtained from vegetative cells without nucleic acid extraction by Dugan and colleagues [102] in *Bacillus anthracis* vegetative cells and spores. The researchers assessed the sensitivity of this procedure to 30 colony-forming units (CFUs) per reaction. This improvement minimised sample processing and the use of the Eriochrome Black T (EBT) dye allowed Etchebarne and colleagues to prepare an infection diagnosis panel (In-Dx) based on the LAMP method [84]. Pre-treatment of collected samples from skin infections includes heat lysis at 95 °C for 15 min and mixing with isothermal reaction reagents. Isothermal amplification, visualized via the presence of the EBT colour, only takes 35 min, resulting in a 1 h detection of 15 microbial pathogens, e.g., *Escherichia coli*, *Staphylococcus aureus*, *Enterococcus faecalis*, *Klebsiella pneumoniae*, *Staphylococcus epidermidis*, *Streptococcus pyogenes*, *Proteus mirabilis*, *Pseudomonas aeruginosa*, *Candida albicans,* or *Enterococcus casseliflavus.* This multiplex detection was previously presented by Oh and colleagues [103], who evaluated a method for the detection of food-borne bacteria. Those data suggest that multiple organisms can be detected in one isothermal reaction with the use of a cocktail of isothermal reagents, primers, and DNA templates.

Loop-mediated isothermal amplification of DNA could be easily developed to diagnose cutaneous melioidosis, an important medical issue. It can present as an ulcer, pustule, or as crusted erythematous lesions caused by infection with *Burkholderia pseudomallei* [104]. Case reports suggest that this type of infection requires a differential diagnosis of nodular or ulcerative cutaneous lesions because typical skin lesion antibiotics do not provide adequate coverage for melioidosis [104]. Chantratita and colleagues used LAMP to detect *B. pseudomallei*, targeting the TTS1 gene cluster [83]. A positive LAMP reaction was visualized by the presence of turbidity, which can be determined by the naked eye. The LAMP method is moderately sensitive (66.7%) compared to real-time PCR (33.3%) for swab sampling; both methods exhibit high specificity (100%). For the rapid diagnosis of melioidosis, it was concluded that the diagnostic overall sensitivity of both assays was low in this evaluation, but the timing of sampling likely proved critical [83]. Very good sensitivity and specificity were achieved in studies by Lim and colleagues on the detection of *Staphylococcus aureus* with LAMP [105]. Furthermore, high sensitivity of isothermal amplification of microbial DNA was shown in a rapid detection system for *Listeria monocytogenes*, where Tang and colleagues compared it to PCR [106]; LAMP exhibited a 100-fold higher sensitivity. Other studies have shown that loop-mediated isothermal amplification of DNA can be used for detection of Gram-positive microorganisms, including *Streptococcus agalactiae* [107], *Streptococcus dysgalactiae, Streptococcus uberis* [82], and Gram-negative *Actinobacillus actinomycetemcomitans* [101], *Escherichia coli* [108], *Klebsiella pneumoniae* [109], or *Coxiella burnetii* [100]. Other studies showed the usefulness of rapid detection of a new class of beta-lactams with the LAMP method [110]. The authors suggested that a newly identified metallo-β-lactamase gene discovered by their group and isolated from an *Alcaligenes faecalis* plasmid was a gene of interest for trail sensitivity. Their results indicate that the LAMP method was proven to be fast, sensitive, and specific for detection of the blaAFM-1 gene.

The LAMP technique is a rapid and sensitive method for microbial detection, and further evaluation of this technique can facilitate the diagnosis of chronic wound infections. The key step in LAMP is proper primer design to obtain good sensitivity, specificity, and efficiency of the reaction. To date, many sets of primers have been designed for microbial detection and evaluated for rapid detection. Here, we present a list of previously described sets of primers that have been shown to be useful in microbial detection (Table 1).

In summary, the presented techniques facilitate the diagnosis of pathogens in chronic wounds. The typing of bacteria causing chronic wound infection allows for assessments of the relationships between bacterial strains involved in different patients and times of infection. The use of qPCR provides the ability to track changes in the expression of selected bacterial and human genes during infection, allowing for treatment modification. Isothermal amplification of DNA is currently the easiest and most economical method for microbial detection in chronic wound infection. Direct visualization of the reaction with dyes and omitting DNA isolation have increased the potential use of this method. The most promising advantages of the LAMP method are its simplicity of implementation compared to cultivation methods and genotyping based on PCR or sequencing, and the fact that it does not require expensive equipment, while offering sensitivity and specificity similar to qPCR. LAMP, compared to cultivation methods, does not require waiting time for the result, often counted in days. Compared to qPCR, LAMP offers comparable or higher sensitivity and specificity while reducing the number of steps, equipment, reagents, and costs. The most important disadvantages of LAMP are the difficulty in designing appropriate primers that recognize the target gene and its high sensitivity to false positive results.

## Figures and Tables

**Figure 1 ijms-25-00411-f001:**
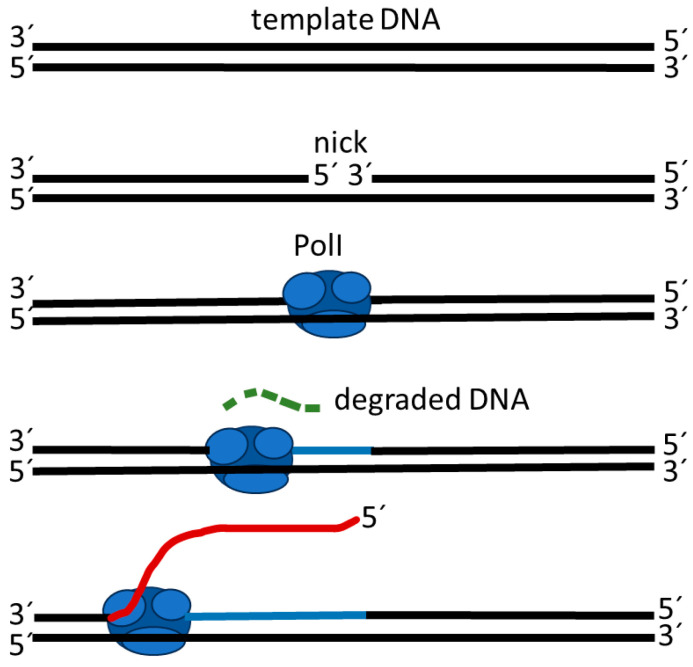
Strand displacement activity of DNA polymerase I (PolI), according to Giannattasio and Branzei [90].

**Figure 2 ijms-25-00411-f002:**
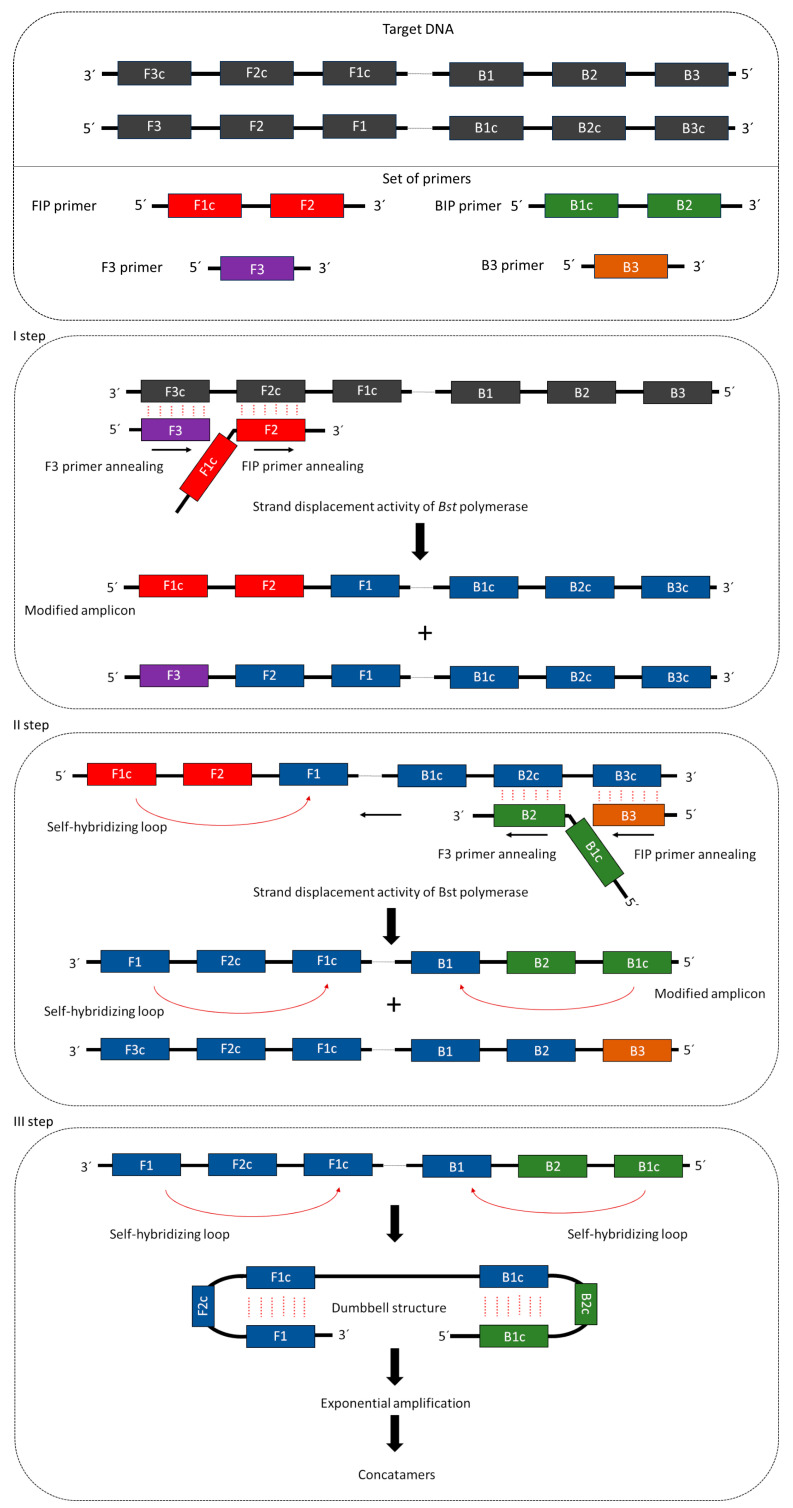
Main steps of LAMP amplification. A set of primers and DNA template with target genre are listed in the first section. In I step a FIP and F3 primers recognize and bind to the template. Bst polymerase with strand displacement activity elongate the DNA strand, where two types of amplicons are synthesized, according to FIP and F3 primer sequences. In the II step, an amplicon synthesized with FIP primers stands as a DNA template for next reaction with BIP and B3 primers. The second modified amplicon is synthesized in this step, where sites homologs to F1 and B1, denoted as F1c and B1c, are added to both reactions. These added sites which proceed to the self-hybridization step (III step), resulting in a dumbbell structure formation. Exponential amplification of such a structure results in the amplification of DNA in concatamers. The figure was prepared based on the data and figures presented in [98].

**Table 1 ijms-25-00411-t001:** Sets of genus-specific LAMP primers developed for rapid microorganism detection. Isothermal amplification of target product may be preceded by incubation at 65 °C for 30 min, followed by heat inactivation at 80 °C for 20 min, depending on the enzyme used in procedure.

Species	Primers	Sequence (5′-3′)	References
*Pseudomonas* *aeruginosa*	F3	CAAGCGCAAGATAGTCGCC	[111]
	B3	TCCGCTTGAACAGGCTGGTG	
	FIP	GAAGATATCCGGCTGGTTGCTTTTCAAGAGGGAATGCCGCAGT	
	BIP	AACGGATCATCGGCATCCTGGTTTTCATCGCCGTCCACAGGTAGA	
*Acinetobacter* *baumannii*	F3	CACAACAAGTTGTTCTTCATAGAT	[112]
	B3	CGAACTCCTGACCTCCTA	
	FIP	AGACTTGAACTTGTGACCCCACTGAGGGTCTGTAGCTCAG	
	BIP	ACCATGACTTTGACTGGTTRAAGTTCGCTCTACCAACTAAGCTAAG	
*Staphylococcus* *aureus*	F3	TCGCTTGCTATGATTGTGG	[113,114]
	B3	ACATACGCCAATGTTCTACC	
	FIP	GTACAGTTTCATGATTCGTCCCGCCATCATTATTGTAGGTGT	
	BIP	TGTTCAAAGAGTTGTGGATGGTGTACAGGCGTATTCGGTT	
	FLP	TTGAAAGGACCCGTATGATTCA	
	BLP	GATACGCCAGAAACGGTGA	
*Proteus* *mirabilis*	F3	AAAAAACGCGGWTCTGCA	[115]
	B3	AAGACAGATAGAGCCAACG	
	FIP	CTGTCGAGCTATGGGTATTAATCACTTTTATTGCGTAATTGGTTAAAARTC	
	BIP	GTTAGTTGCGCTATCTTGTGCTTCTTTTGAACGTGATACATCGGTAGA	
	LF	CCGCCATAGTACGTACTCGCCA	
*E. coli*	F3	GCC ATC TCC TGA TGACG	[116]
	B3	ATT TAC CGC AGC CAG ACG	
	FIP	CTG GGG CGA GGT CGT GGT ATT CCG ACA AAC ACC ACG AATT	
	BIP	CAT TTT GCA GCT GTA CGC TCG CAG CCC ATC ATG AAT GT	
	LF	CTT TGT AAC AAC CTG TCA TCG ACA	
	LB	ATC AAT CTC GAT ATC CAT GAA GGT G	
*K. pneumoniae*	F3	GGA TAT CTG ACC AGT CGG	[117]
	B3	GGG TTT TGC GTA ATG ATC TG	
	FIP	CGA CGT ACA GTG TTT CTG CAG TTT TAA AAA ACA GGA AAT CGT TGAGG	
	BIP	CGG CGG TGG TGT TTC TGA ATT TTG CGA ATA ATG CCA TTA CTT TC	
	LB	GAA GAC TGT TTC GTG CAT GATGA	
*E. faecalis*	F3	GCC GGA AAT CGA TGA AGA	[118]
	B3	TCC AGC AAC GTT GAT TGT	
	FIP	CAC TTT TTG TTG TTG GTT TTC GCT TTA TTA TCT GCT TGG GGT GC	
	BIP	ATC TGC AGA CAA AGT AGT AAT TGC TCC AAG CTT TTA AGC GTG TC	
	LF	AAA TGC TGC GCC AGC TCG	
	LB	TCC AAT GTG GAA CTT AAA CGT ACC	

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
