# Peer review of "Loop-Mediated Isothermal Amplification of DNA (LAMP) as an Alternative Method for Determining Bacteria in Wound Infections"

_ijms, 2023, doi:10.3390/ijms25010411_

Round 1

Reviewer 1 Report

Comments and Suggestions for Authors

The manuscript “An overview of rapid molecular methods of bacterial determination in wound infection” by Gieron et al, is a review of the various molecular methods for the detection and identification of bacterial infections. The authors lean towards the Loop-mediated isothermal amplification of DNA (LAMP) and offer diagnostic primers for different pathogens. Unfortunately, I think that this manuscript does not offer the novelty and innovation or even clarity that is expected from such a review.

Major concerns:

The focus of the manuscript is rapid molecular methods of bacterial identification in wound infections. From reading the manuscript introduction I can’t understand the specific challenges in wound diagnostics that set them apart from other clinical samples, nasal swabs for example. The authors do not explain why, if we are dealing with chronic wounds, time is of the essence and you need a rapid (minutes) bedside test. And what are the major advantages of nucleic acid-based tests over other strategies which are based on peptides for example. As I understand hospital labs are based mainly on commercial kits rather then in house experimental methods, the challenges in incorporating R&D methods into a clinical lab must be addressed.

The molecular methods described in this manuscript are not cutting edge. rtPCR ad qPCR are considered basic and sequencing based methods that became more popular and less expensive are absent. This becomes significant in section 3, “Genotyping of strains from wound infection”, were methods such as pulsed field gel electrophoresis (PFGE) and PCR-based fingerprinting are described, methods that I doubt are in common use these days.

Section 4, which is the most significant part of this review, is unclear, for example, in order to understand the mechanism of LAMP (4.2), I had to go to the original manuscript (ref 65 for example). Figure 2 is unclear and does not explain the loop formation.

The context of the primer list in table 1 and 2 is unclear since beside the pathogen name and reference no performance data is presented.

In general, I would expect to see in a review like this a side-by-side comparison of the different methods with pros and cons. This may better serve the readers and allow a better understanding of the authors conclusions.

Reviewer 2 Report

Comments and Suggestions for Authors

The manuscript entitled “An overview of rapid molecular methods of bacterial determination in wound infections” is about to show the state of the bacterial diagnostic method in wound area.

Manuscript was written well bit need some modification to be published.

Overall, this manuscript is about to state the bacterial diagnostic method in wound area, but generally authors talk about the methods that used to detect the bacteria, not focus to the wound area. Even though I think authors not include enough example of detection methods. So I think authors need to modify the title or rearrange the manuscript.

At line 48: What does the “planktonic strategy” mean?

At line 81, cannot be uncultured should be cannot be cultured or can be uncultured.

Authors descript the PCR method, even though the description is brief, I think most of researcher who can read this manuscript know what is the PCR. So, authors rather reduce the description about the PCR from line 98 to line 104 and also other methods.

At line 296, 1-h hour should be 1 hour

Authors need to consistently use of reference style.

Round 2

Reviewer 1 Report

Comments and Suggestions for Authors

The revised manuscript of Gieron et al; Loop-Mediated isothermal amplification of DNA (LAMP) as an alternative method for determining bacteria in wound infections” is an improved version of the previous manuscript. Some revisions are needed prior to publication:

The question about the use of R&D kits in clinical labs was not addressed in the manuscript or the corresponding letter.

Line 19 The word “however” is out of place and can be omitted

Line 78 Give numerical value and reference instead of “period”

Lines 82-85 The determination of the “antibiotic therapy” is not performed by susceptibility testing? Please clarify this point.

Line 97 What is the proportion of anaerobic bacteria in wound infections?

Line 195 What is the relevance of “investigating the spread” in wound infections?
